# Plant-Derived Natural Compounds for Tick Pest Control in Livestock and Wildlife: Pragmatism or Utopia?

**DOI:** 10.3390/insects11080490

**Published:** 2020-08-01

**Authors:** Danilo G. Quadros, Tammi L. Johnson, Travis R. Whitney, Jonathan D. Oliver, Adela S. Oliva Chávez

**Affiliations:** 1Texas A&M AgriLife Research, San Angelo, TX 76901, USA; dan.quadros@ag.tamu.edu (D.G.Q.); trwhitney@ag.tamu.edu (T.R.W.); 2Department of Rangelands, Wildlife and Fisheries Management, Texas A&M AgriLife Research, Texas A&M University, Uvalde, TX 78801, USA; Tammi.Johnson@ag.tamu.edu; 3Environmental Health Sciences, School of Public Health, University of Minnesota, Minneapolis, MN 55455, USA; joliver@umn.edu; 4Department of Entomology, Texas A&M University, College Station, TX 77843, USA

**Keywords:** integrated pest management (IPM), natural acaricide, natural repellent, plant extract, tick-borne diseases, tick control

## Abstract

Ticks and tick-borne diseases are a significant economic hindrance for livestock production and a menace to public health. The expansion of tick populations into new areas, the occurrence of acaricide resistance to synthetic chemical treatments, the potentially toxic contamination of food supplies, and the difficulty of applying chemical control in wild-animal populations have created greater interest in developing new tick control alternatives. Plant compounds represent a promising avenue for the discovery of such alternatives. Several plant extracts and secondary metabolites have repellent and acaricidal effects. However, very little is known about their mode of action, and their commercialization is faced with multiple hurdles, from the determination of an adequate formulation to field validation and public availability. Further, the applicability of these compounds to control ticks in wild-animal populations is restrained by inadequate delivery systems that cannot guarantee accurate dosage delivery at the right time to the target animal populations. More work, financial support, and collaboration with regulatory authorities, research groups, and private companies are needed to overcome these obstacles. Here, we review the advancements on known plant-derived natural compounds with acaricidal potential and discuss the road ahead toward the implementation of organic control in managing ticks and tick-borne diseases.

## 1. Introduction

Ticks are the most important vectors of vector-borne diseases in the United States and one of the main arthropod vectors of human and animal pathogens worldwide, representing a substantial economic burden. The impact of Lyme disease alone has been calculated at approximately US$1.3 billion per year [1]. Furthermore, climate change may be affecting the distribution of tick species of human and animal health importance and is likely to extend the transmission cycle of many tick-borne pathogens in the US [2,3]. Similar predictions have been made in Europe concerning the distribution and extended questing activities of *Ixodes* spp. ticks [4,5]. Thus, additional options to control the spread of tick-borne diseases are warranted.

Due to the magnitude of economic losses related to ticks and tick-borne diseases for animals and humans, as well as tick-acquired acaricide resistance, negative environmental impacts, and potential chemical acaricides residues in food, novel methods of tick pest control and prevention based on natural organic molecules have been pursued. The objective of this paper is to highlight the most prominent plant extracts and secondary metabolites for tick control and prevention and challenges to scale up the acceptance and use of these products on a commercial scale. Final remarks will be directed towards the efficacy of natural organic compounds for tick pest control and answer whether their use in livestock production systems and wildlife is pragmatic or utopian. This review focuses only on plant-derived natural compounds and does not evaluate other biological control alternatives such as entomopathogenic fungi or nematodes [6,7]. Reviews related to alternatives for the control of ticks are available [8,9,10,11].

## 2. Ecology and Economic Importance of Ticks and Tick-Borne Diseases of Livestock (Veterinary) and Public Health Concern

Ticks are obligate hematophagous arthropod ectoparasites that entirely depend on one or more hosts (e.g., mammals, birds, or reptiles) to complete their life cycle. More than 900 species of ticks exist globally. They are found in subarctic to Antarctic regions and in habitats that range from rainforest to deserts [12]. Hard ticks (Ixodidae) can be classified in one-, two-, or three-host ticks based on the number of hosts they utilize throughout their lifespan. One-host ticks seek out a host at the larval life stage and remain on the same host for the subsequent life stages until they drop off to lay eggs. As the name suggests, two- and three-host ticks obtain blood meals from two or three different individual hosts, respectively. Two-host ticks acquire blood meals as larvae and nymphs from a single host before dropping off the host to molt to the adult stage and then seeking out a second host. Three-host ticks drop into the environment and molt to the next stage after each blood meal. They must relocate to another host to obtain a subsequent blood meal. Thus, three-host ticks are often less host specific and frequently feed on small mammals, including birds and rodents, as immature stages and larger animals as adults. Three-host ticks normally have a lifespan of two to three years.

Several prominent tick-borne diseases of veterinary and public health importance are transmitted by ticks that can utilize both domestic and wild animals as hosts (Table 1).

As nearly 100% of the tick-borne pathogens present in the United States have a potential wildlife host component, a need remains to provide new economically feasible tools to reduce contact at the livestock–wildlife interface [33]. The development of strategies that decrease pathogen transmission between wildlife, livestock, companion animals, and human beings is required [33,34,35]. Therefore, the research and implementation of integrated control and prevention methods for ticks and tick-borne diseases are necessary to reduce ecological and economic impacts. Representative tick-borne diseases of veterinary importance and public health importance, associated vectors, and causative agents are described in Table 1.

The direct and indirect economic impacts related to ticks and tick-borne diseases are significant. Roughly 80% of the world’s cattle population is at risk of tick infestation and tick-borne diseases, which account for economic losses estimated up to US$30 billion yearly [36]. In addition to pathogen transmission, the infestation of animals with ticks causes other problems. When ticks bite their hosts, skin tissue injury occurs, which includes irritation, inflammation, or hypersensitivity [37]. The lesions also predispose the animal host to dermatitis, secondary bacterial infections, or myiasis [38]. In consequence, the animal becomes stressed, which affects behavior, production, and welfare [39]. Tick bites directly depreciate the quality of hides and skins and, consequently, the value of leather because they can become hard, opaque, perforated, and rough [40,41].

As hematophagous arthropods, ticks rely on blood as their only source of nutrients and can consume several hundred times their unfed body weight in blood [42,43]. In this way, ticks can cause anemia; immunosuppression; decreased feed intake, digestion, and metabolism; reduced milk production and quality; reduced weight gain and body condition; and reduced reproduction (e.g., increasing abortion rates and lowering pregnancy rates). Infection can become significant enough to cause death in some animals [44,45,46,47].

Tick-borne pathogens are transmitted through tick saliva, a phenomenon called saliva-assisted transmission [48]. Tick saliva can also contain a neurotoxin that causes host paralysis [49,50,51]. Thus, ticks and tick-borne diseases affect animal and human health worldwide.

## 3. Chemical Control Failure and the Hope of Natural Organic Products on Tick Pest Management

The integration of multiple strategies or integrated pest management (IPM) for tick control is considered the ideal approach. These strategies may include, but are not limited to: development of host resistance, pasture management, IPM based on tick abundance, biological control (e.g., ants, predatory mites, chickens, and others), the use of vaccines against ticks or tick-borne diseases (when available), acaricide resistance management, and cost/benefit analyses of acaricidal application [36,52,53,54,55,56,57,58,59,60,61].

The use of synthetic acaricides is the most widely implemented method that producers use to control ticks [62,63,64,65]. In 2019, the estimated worldwide market value of acaricides was US$275.1 million. The animal husbandry industry alone accounts for roughly one-quarter of this market, while the remainder is represented by crop defense, home applications, and other uses [66]. The overuse and misuse of acaricides has led to acaricide resistance in some tick populations [39,62,64,67,68,69]. The number of reports of tick resistance to synthetic acaricides around the world is alarming (Table 2). The development of resistance to a new acaricide can now be expected within five to 10 years of its introduction unless practices are changed and resistance management is implemented [64]. When trying to overcome the problem of resistance, livestock managers often increase the frequency of application and recommended dose, mix products, and use “off-label” products, which contribute to the surge of tick populations with multiple resistance traits [64,65,70]. Further, in some cases acaricide resistant tick populations have been detected in wildlife populations. In Texas, permethrin resistant ticks with two mutations to the voltage-sensitive sodium channel (*Vssc*) gene have been collected from white-tailed deer and nilgai [71].

Beyond the resistance dilemma, the use of acaricides can have harmful effects on animals, humans, and the environment. A recent study detected residues of several pesticides in 26% to 60% of the milk samples collected from conventional dairy farms, but no residues were observed in organic milk samples [106]. In developing countries, acute pesticide poisoning due to lack of safety precautions has become a problem in terms of public, occupational, and environmental health [107,108]. From an environmental perspective, acaricides can have multiple, wide-ranging effects: organochlorines can persist in soil and are highly toxic to many arthropods; organophosphates are less persistent than the organochlorines in the land, but generally have much higher toxicity to birds and other wildlife; pyrethroids are toxic to fish and aquatic organisms as well as non-target and beneficial arthropod species; and carbamates tend to be more persistent in soil and vary significantly in mammalian toxicity [109].

A shift from conventional synthetically derived acaricides to more sustainable and naturally based organic control options is needed [84,110]. Novel tick control options can be incorporated into an IPM plan to decrease the risk of acaricides on public health and the environment [46,60,84,107,110]. Knowledge about local and regional plants with acaricidal properties is now necessary for developing safe, efficient, affordable, accessible, environmentally friendly, and community-driven successful strategic interventions for tick control and management programs [110].

The use of organic compounds for safe and efficient tick control may be suitable for organic and conventional livestock production systems. Both systems will strongly benefit from the development of commercial products based on natural compounds. Organic livestock production is experiencing rapid expansion. Nevertheless, organic farmers have a very limited number of options for tick control. Global organic production has increased over recent years as a result of growing consumer demand and public concern for sustainability [111,112]. In the US, “organics” are the fastest-growing segment of national agriculture. Dairy products represent a substantial portion of the market (15%), while meat/fish/poultry accounts for just 3% [113]. In the European Union, the portion of organic livestock remains small, despite the substantial increase in organic meat consumption in the last decade. Organic livestock systems represent only 0.7% and 3.3% of swine and poultry production, respectively, whereas sheep and bovines represent just 5.0% and 5.2% of the organic livestock production, respectively [111].

The current state of natural plant-based acaricides and tick repellents already available in the market is unknown, but this area has seen a sharp increase in interest from researchers, government, industry, and the public. Plants with insecticidal effects can be a promising alternative, with reduced toxicity to mammals, biodegradable characteristics, and less chance of development of resistance. Therefore, research focusing on acaricidal substances or repellents of plant origin should be further encouraged [114,115].

## 4. Plant-Derived Compounds with Potential Use for Tick Pest Control

Globally, more than 200 plant species with tick-repellent or acaricidal properties are known [116]. Essential oils, extracts, or pure allelochemicals are the primary forms of plant-based products with biocidal features in tick assays. Methods including steam distillation [117], hydrodistillation [118], ethanolic and aqueous extraction [119], methanolic extraction and spilanthol [120], and hexane, ethyl, and acetate extractions [120,121] have been used to obtain these substances.

Plant essential oils, which are the most studied plant-derived compounds for tick control and prevention [114,115,122,123], are complex mixtures of natural, volatile organic compounds predominantly composed of terpenic hydrocarbons [124]. Depending on how these compounds are extracted, they can be classified into different groups with variations in efficacy. A list of organic compounds with acaricidal and tick repellence properties is presented in Table 3.

Despite the fact that medicinal plants have been used in ethnoveterinary and human medicine dating back to ancient times, some can have toxic properties depending on their origin and nature [129,130]. Toxic plants, however, may contain active compounds with useful biological activities for biomedical applications [131,132]. For instance, glycosides, alkaloids, saponins, tannins, volatile oils, flavonoids, and diterpenoids are examples of active components that can be potentially toxic but are used in ethnoveterinary applications [133,134]. Obviously, the intercalating concern between pharmacology and toxicology is dose-dependent [131]. It is essential to be aware of the toxicity that plants of veterinary significance can have to avoid disease or mortality in livestock [135,136]. Studies defining the concentration and dosage of specific plant components or extracts that lead to detrimental effects in animals and humans are needed to define working dosages.

### 4.1. Plant Extracts

The preparation of plant extracts involves the isolation of bioactive compounds from plants with solvents and processes that may include maceration, heat extraction, microwave-assisted extraction, sonication, and other methods [137].

Extracts from *Acmella oleracea* (Asteraceae), the jambu plant of Amazonia, have been evaluated extensively for acaricidal activity on a variety of tick species and life stages. Extracts from this plant using hexane [138] and methanol [120,139] have proven more effective than aqueous, ethanol [140], and chloroform extractions [141]. The methanol extract is particularly toxic to tick larvae with an LC_90_ of 1.6 mg/mL and 6.6 mg/mL for larvae of *R. microplus* and *Anocentor nitens*, respectively [120,139]. It is also effective against nymphal and adult ticks, even though greater concentrations are required to achieve LC_90_. The lethal effects of *A. oleracea* are often attributed to the presence of the fatty acid amide spilanthol in the plant, and toxicity studies of the isolated constituent have demonstrated its efficacy [120]. However, *A. oleracea* specimens with low concentrations of spilanthol also demonstrate high lethality against ticks, indicating the presence of other effective compounds or synergistic effects [139].

Extracts from *Annona muricata* (sweetsop) and *Annona squamosa* (soursop) plants also exhibit acaricidal activity. A 2% solution of *A. muricata* seeds extracted with ethanol killed 100% of engorged *R. microplus* females [142]. Studies using *A. squamosa* extracts found these to be more effective with aqueous extracts of fruit peels, demonstrating an LC_50_ of 0.405 mg/mL for *Haemaphysalis bispinosa* adults and an LC_50_ of 0.548 mg/mL for *R. microplus* larvae [143]. Hexane-extracted leaf material proved even more toxic with an LC_50_ of 0.145 mg/mL against *Ha. bispinosa* adults [143].

Extracts from *Nicotiana tabacum*, the cultivated tobacco plant, are also potently acaricidal. While methanol extractions required greater concentrations (25–100 mg/mL) to kill 50% of *Rhipicephalus decoloratus*, *Rhipicephalus pulchellus*, or *R. sanguineus* adults within 24 h of exposure [144,145], hexane extractions proved more effective with an LC_50_ of 0.6 mg/mL for female *R. microplus*. Ethanol extracts also appeared to produce a greater level of toxicity [146] than other extracts.

Neem has also been tested as an acaricide. Aqueous extracts and oils have been tested due to their potential use as insecticides [147] and acaricides [148]. Azadirachtin is the most widely studied component purified from neem oils. A study comparing the effects of different concetrations of Azadirachtin and neem leaf extracts on *R. sanguinneus* larvae showed 80% and 95% mortalities after the Larval Packet Test (LPT), respectively, in the lower concentrations (0.5% and 10%, respectively) [148]. Nevertheless, this acaricidal effect was diminished in experiments with higher concentrations [148]. In adult females ticks, treatment with the aqueous extracts from neem leaf at 10% and 20% results in morphological changes during oocyte development when compared to control samples [149]. Similar outcomes were reported after treatment with neem oils containing Azadirachtin [150], indicating that the changes in morphology are possibly connected with this compound. Furthermore, neem oil also reduces cuticle thickness and distorts epithelial cell morphology of semi-engorged females [151]. However, neem leaf extracts can negatively affect oocytes and ovaries in mammals. Studies with rats indicate that neem leaf extracts increase oxidation in the oocytes and lead to apoptosis [152,153]. Although neem leaf material can produce anemia, reduced fertility, and cause abortions, aqueous extracts and purified components appear to be less toxic and require high concentrations to have negative effects on mammals [147], supporting the potential of this plant as a source for acaricidal compounds.

Tannins (phenolic compounds of high molecular weight ranging from 500 to more than 3000) are the most abundant secondary metabolites synthesized by plants [154]. Hydrolysable tannins (HTs) and condensed tannins (CTs, also known as proanthocyanidins) are two major classes of secondary metabolites that form an important line of defense against herbivory [155]. HTs may be toxic to livestock, including ruminants, while CTs can have anti-nutritional effects when animals consume high concentrations of biologically active forms [156]. However, tannins possess various biological activities including antimicrobial, anti-parasitic, anti-viral, antioxidant, anti-inflammatory, and immunomodulation [157]. Therefore, they have drawn attention from many research groups. Fresh and dry *Aloe arborescens* extracts prepared using various solvents (pure ethanol, ethanol-dichloromethane binary mixture, and ethanol-dichloromethane-acetone ternary mixture) containing water-soluble tannins show that tannins had a strong effect on the number of eggs laid and larval hatching rate of *R. microplus* [158]. Trials with CTs are more common. Four tannin-rich plant extracts (*Acacia pennatula, Piscidia piscipula, Leucaena leucocephala,* and *Lysiloma latisiliquum*) showed acaricidal effects against larvae of *R. microplus* (54.8%, 88.14%, 66.79%, and 56.0%, respectively), but no effect on adults or egg-laying [84]. Ethanolic extracts prepared from the leaves of *Schinopsis brasiliensis* (CTs 0.36%), *Piptadenia viridifora* (CTs 1.01%), *Ximenia americana* (CTs 0.35%), and *Serjania lethalis* (CTs 3.37%) at 25 to 150 mg/mL were tested for *A. nitens* control [158,159]. Extracts of *X. americana* and *P. viridifora* showed effective inhibition of tick reproductive parameters (LC_90_ 78.9 and 78.9 mg/mL, respectively), even though these plant species have low CT concentrations. This indicated that components other than CTs (e.g., flavonoids) were involved in *A. nitens* control.

More recently, six medicinal plants (*Vernonia amygdalina*, *Calpurnia aurea*, *Schinus molle*, *Ricinus communis*, *Croton macrostachyus*, and *Nicotiana tabacum*) were used against *R. decoloratus* and *R. pulchellus* in an in vitro adult immersion test with five crude extract concentrations (6.25, 12.5, 25, 50, or 100 mg/mL) with diazinon (0.1%) as a positive control. Two of the concentrations (50 and 100 mg/mL) showed a comparable and strong acaricidal effect on both tick species compared with the positive control after a 24 h period [144].

Even though plant extracts are promising for tick control, many secondary compounds may be responsible for the acaricidal effect of these extracts (i.e., terpenes, stilbenes, coumarins, acids, alcohols, sulfide compounds, tannins, and aldehydes) [160]. More studies are necessary to isolate and test specific compounds in bioassays with ticks in the larval and adult stages to determine their safety to humans and other animals.

### 4.2. Plant Essential Oils

Essential oils are obtained from plant extracts and are separated from the aqueous phase through conventional methods such as distillation, hydrodiffusion, and solvent extraction [161]. Several essential oils have been tested in different formulations and against different tick species. Treatment with essential oils from some plants can affect tick mortality, fecundity, and egg hatching rate [122]. Recent studies examined the potential use of essential oils of *Laurus nobilis* (laurel) [162], *Ocotea odorifera* (canela sassafras) [163], *Chrysopogon zizanioides* [164], and *Schinus molle* [165] with levels of control between 7.59% to 99%, depending on the concentration of essential oil, tick life stage, and species. The combination of essential oils appears to enhance the effect of individual oils [166,167]. A promising blend of essential oils is *Cinnamomum verum* (cinnamon), *Cuminum cyminum* (cumin), and *Pimenta diocia* (allspice), which achieved acaricidal activity from 90% to 100% [165]. Essential oils can also be used as repellents on clothing and on companion animals for tick species such as *I. ricinus* [168,169], highlighting the potential of plant-derived compounds to be used in multiple ways to aid in the control of ticks of veterinary and public health importance.

Nonetheless, due to the variations in essential oil extracts, scientists are now interested in identifying the specific component(s) with acaricidal properties in these plant extracts. Ferreira et al. [170] identified eugenol as the main constituent of clove’s essential oil. Both clove essential oil and eugenol resulted in significantly reduced egg production index (EPI) and hatching. The acaricidal effect of eugenol on *R. sanguineus* unfed larvae, unfed nymphs, fed larvae, and fed nymphs was demonstrated by a later study [171]. More significant mortalities were achieved when eugenol and thymol were combined. Furthermore, this formulation reduced application costs to $3.96/L [171].

Many studies have examined the efficacy of non-seed oil plant extracts as acaricides, particularly those with demonstrated insecticidal or antiparasitic effects, or those with a traditional history of medicinal usage. Summaries of these studies are reviewed in Benelli et al. [115] and Rosado-Aguilar et al. [172].

### 4.3. Mode of Action

The mode of action of many plant-derived compounds used for tick control is not completely understood. Some essential oils cause neurotoxic effects, such as inhibition of acetylcholinesterase (AChE), antagonism with the receptors of the octopamine neurotransmitter, or closure of the chloride channels by gamma-aminobutyric acid (GABA) [173]. Likewise, the exact mode of action of many plant essential oils has not been elucidated, and few studies have been conducted to understand how these naturally occurring compounds act on ticks. Gross et al. [174] developed an in vitro assay, using hamster cell lines (CHO cells) expressing the *R. microplus* Tyramine receptor RmTAR1e, to screen components from essential oils that interact with this receptor. Using this assay, they identified a Tyramine receptor as the potential target for pulegone, carvacrol, isoeugenol, 1,4-cineole, and piperonyl alcohol. However, how other components and essential oils affect tick biology remains unexplored and warrants further investigation.

## 5. From the Bench to the Market: A Long Rough Road of Scaling up Natural Products for Tick Pest Control

Acaricides are pesticides designed specifically for mites and ticks. Natural compounds of botanical origin are increasingly being investigated to develop novel biopesticides for agriculture [122,175]. The growth of biopesticides is projected to outpace the synthetic pesticides, with annual growth rates of between 10% and 20% [176].

The identification of active ingredients in natural products with acaricidal and repellent properties is currently a prominent area of investigation [114]. There are two approaches used to identify plant-derived natural products for tick pest control and prevention. The first method is based on traditional ethnobotanical knowledge and the familiarity with regional sources, such as crude plant extracts that contain a blend of different plant metabolites, which can be used against ticks [116,177]. In this approach, scientific validation should be conducted to expand the applicability of “homemade” recipes. Further, these recipes should be disseminated to the public by rural extension services, as this method may be more applicable to smallholder farmers in developing countries. The second approach includes commercially manufactured products. In the past, these products were usually produced by small companies of local importance [178]. Nowadays, large agrochemical companies have become more involved through the in-licensing of technology and products, joint ventures, and acquisitions [176]. Although numerous products are in the market today, they are usually developed based on active substances obtained only from a few plant species [178]. The current challenges are to identify and produce these compounds on a large scale to meet the growing demand.

Regulatory approval remains the final and often most difficult barrier to overcome in commercializing a pesticide. Some plant-based formulations are on the ‘‘Generally Recognized as Safe’’ (GRAS) list of the U.S. Food and Drug Administration (FDA) [122]. Minimum Risk Exemption regulations in 40 CFR 152.25 exempted a list of active ingredients, including, among others, geraniol, eugenol, and citronella, from the Federal Insecticide, Fungicide, and Rodenticide Act (FIFRA) [179]. Fewer biopesticide-active substances are registered in the European Union (EU) than in the United States, India, Brazil, or China, due to long and complex registration processes in the EU and other countries, which follow the model for the registration for conventional pesticides [173,180]. In these cases, the approval is based on a review of data on product chemistry, environmental fate, and toxicology to laboratory animals and non-target organisms, including fish, wildlife, and pollinators, while efficacy data is required for some agencies [181].

In the United States, regulatory requirements imposed by the responsible agencies have slowed the introduction of new biopesticides [164]. Moreover, there has been confusion regarding the Food and Drug Administration (FDA) and the Environmental Protection Agency (EPA) jurisdiction over natural organic acaricides. The EPA regulates biotechnology-based pesticides, while the FDA regulates the safety of human and animal foods. To modernize the regulatory framework for biotechnology products, the EPA recently released “The Unified Website for Biotechnology Regulation” in coordination with the U.S. Department of Agriculture (USDA) and the FDA, which provides information about actions the federal government is taking to oversee development of agricultural biotechnology products [182].

Other important general issues that can affect the development of commercial natural organic biopesticides are (a) scarcity of the natural resource, (b) the need for chemical standardization and quality control, and (c) other challenges including long-term stability, storage, and transportation [173]. Additionally, product efficacy is inherently a customer expectation. The reduction of efficiency of plant extracts, when tested on animals, is undoubtedly a constraint to the development of alternative acaricides [183]. Many factors can affect the acaricidal activity of an extract including solvent, extraction time, extract concentration, extracted plant age, tick species, and exposure time [177].

Additionally, many promising plant-derived compounds for tick control and repellence are from aromatic plants [184,185]. The distinct odors may cause different reactions depending on the animal species [186]. Livestock and some wild animals are more sensible to odors than humans due to the mechanisms for odor perception [187,188]. In the case of negative response of animals to specific plant components, it may be necessary to restrain their movement to apply the product. Thus, these products will be more applicable to livestock than wild animals.

Nanoparticle and nanoemulsion formulations can enhance the activity and efficacy of biopesticides [181]. Green synthesized nanoparticles using plant extracts are easy to prepare, eco-friendly, cost-effective, and promising in the control of ticks [181,189]. Encapsulation into nanosystems helps overcome some hurdles related to physicochemical properties (e.g., limited stability and handling), enhancing the overall efficacy. Among different nanosystems, micro- and nanoemulsions are easy-to-use systems in terms of preparation and industrial scale-up [190,191].

### Organic Compounds Already in the Market

Some organic compounds are already in the market and their effect on ticks has been widely studied. Here, we will summarize some of the studies that have been done in two constituents from essential oils available in the market: nootkatone and carvacrol. The potential of nootkatone and carvacrol as alternative options to chemical control of ticks was reported by the Center for Disease Control and Prevention (CDC) [192].

Products containing nootkatone, such as NootkaShield [193], are already commercially available. The acaricidal effect of nootkatone has been tested in several tick species [194]. *R. sanguineus*, *I. scapularis*, *D. variabilis*, and *A. americanum* were all susceptible to nootkatone during bioassays, although *A. americanum* has a higher LC_90_ of 0.485 μg/cm^2^. This organic compound also has 100% repellency to *I. scapularis* adults and over 89% to *A. americanum* through three days and seven days, respectively, when applied to coveralls [195]. An encapsulated formulation of nootkatone was developed by Behle et al. [196] to overcome problems with volatilization and phytotoxicity in emulsified formulations. Although both formulations resulted in leaf damage, this was significantly reduced in the encapsulated formulation. Further, the encapsulated nootkatone resulted in higher *I. scapularis* nymph mortality than the emulsified formulation [196]. Plot trials in areas with *I. scapularis* showed that nootkatone can significantly diminish host-seeking ticks for up to 16 days when applied in an emulsifiable formulation [197]. An aqueous application resulted in 100% control of *I. scapularis* and *A. americanum* for up to 21 days after two applications [198]. The encapsulated formulation showed 100% control up to 27 days post-application. However, tick numbers were not significantly different in the following season between treated and untreated trials, granting the need for additional studies.

Several essential oils are commercialized as repellents for pets and humans. However, few have applicability in the field. Carvacrol is a monoterpene purified from essential oils with acaricidal properties. It decreases egg hatching in *R. microplus* ticks [199], probably by negatively affecting oogenesis [200]. Defects in oocyte development were also observed after sublethal treatment of *R. sanguineus* with acetylcarvacrol. Oocytes presented fragmented yolk granules, low protein content, chorion detachment, and vacuolation around the nucleus [201]. A field application of carvacrol resulted in 93% control of *I. scapularis* for 35 days after two applications, but control was reduced to 78% by day 42 [198]. A control level of 62% was achieved for *A. americanum*, but unlike nootkatone, carvacrol is not a good repellent for *A. americanum,* with less than 68% repellency [195]. Nevertheless, this lower repellency may be due to the formulation. Lima et al. [202] demonstrated that encapsulated carvacrol provided control with RC_50_ of 0.05 mg/cm^2^ for *R. microplus* at 6 h post-treatment versus 0.27 mg/cm^2^ for the non-encapsulated carvacrol.

## 6. Final Remarks

Undoubtedly, natural plant-derived compounds are promising tools for an IPM program due to the acaricidal and repellency effects on ticks. The proven efficacy of several compounds and reduced risks for humans and the environment, as well as the industry interest, make this field of research highly necessary. With advances in research, several commercial products based on phytocompounds have been marketed. However, beyond regulatory challenges, standardized methods of organic compound extraction and tick assays are still lacking. More investment and funding are necessary to cover the complexity of field tests involving livestock–wildlife interactions. Chemical standardization needs to be established. Studies evaluating toxicity to non-target organisms and synergistic/antagonistic effects of botanical compounds are necessary.

Although many tick-borne diseases do not pose serious threats to wildlife populations, wildlife can be important to the maintenance and propagation of ticks (infected ticks), which may jeopardize livestock and human health. Tick control in wildlife populations is rarely attempted and presents numerous challenges. When compared to livestock, which are handled and individually treated, treatment of wildlife is more problematic. Many tick vectors utilize a wide range of wildlife as hosts, and these hosts are often spread across broad geographic regions. Therefore, it is often not practical to treat for ticks on this scale. Methods employed for the treatment of wildlife have included the feeding of medicated feedstuffs and topical applications of acaricides [203,204,205,206,207,208]. These methods have been applied with mixed success and all could have broad-reaching unintentional ecological implications. Target treatment is often intractable and some drawbacks to these methods include exposure of non-target species, environmental contamination, and possible human exposure through residues contaminating tissues consumed by hunters.

A potential avenue for the control of ticks in livestock and wild populations is supplementing feed with plant-derived compounds. Although the effect of adding plant-derived compounds to animal feed for tick control is not well tested yet, many plant extracts have shown antiparasitic effects. These compounds can control endoparasites and ectoparasites. One well-known example is avermectins, which affect reproduction, feeding, and motility in parasitic nematodes and hematophagous arthropods [209]. Supplementing animal feed with plant extracts can control endoparasitic infestations. Redberry juniper significantly reduced the number of eggs of the nematode *Haemonchus* spp. during 28 days of feeding and complemented the effect of Ivermectin from days 32 to 42 [209,210]. *Juniperus pinchotii, J. ashei*, *J. monosperma*, and *J. virginiana* have been used as fiber ingredients in lamb diets [211,212]. Supplementing feed with plant extracts or plants to control internal parasites, such as helminths in sheep, goats, and deer is also well documented [210,213]. This antiparasitic effect may also include external parasites like ticks. The acaricidal effect of *Juniper* spp. is suspected to be connected to terpenes [127]. In vitro studies suggest that terpenoids from *Ocotea aciphylla* inhibit acetylcholinesterase [121,127]. Several compounds and essential oils purified from several *Juniperus* spp. have shown acaricidal and repellent properties [126,214,215,216]. Whether enough concentration of the compounds with acaricidal properties makes it to the bloodstream and whether ticks could be controlled by feeding *Juniper* to livestock and wildlife is unknown. Nevertheless, it is tempting to speculate that animal feeds may have a synergistic effect with other acaricides in the market and could enhance tick control, facilitating delivery to wild tick reservoirs. Such a result was achieved in the control of helminths when redberry juniper was complemented with ivermectin [210].

Although generally considered safe for mammals, some plant-derived products have been shown to exert negative health and welfare effects in humans and other animals [210,217]. Therefore, regulations requiring ecotoxicity studies, possible collateral effects, and education about how to use these compounds and the risks that they impose may minimize the possible negative impacts.

As tick distribution and the spread of tick-borne disease increases, the development of alternative methods and products to control tick populations should become a priority for funding agencies and the industry. Likewise, field tests that evaluate the efficacy of existing products for their ability to control tick populations in livestock and wild-animal populations are imperative. By increasing the arsenal of products that can be safely used in livestock and wild animals, we can reduce the concerns regarding acaricide impact on human health, the development of acaricide resistance, and may even be able to reduce the impact of tick-borne diseases worldwide.

## Figures and Tables

**Table 1 insects-11-00490-t001:** Representative ticks and associated tick-borne diseases of human, livestock, and wildlife, and their causative agents and hosts.

Primary Tick Vector	Disease	Causative Agent	Host(s)	Reference
*Amblyomma americanum*	Ehrlichiosis	*Ehrlichia chaffeensis*, *E. ewingii*	Humans, dogs	[13]
*A. mixtum* *	Equine Piroplasmosis	*Theileria equi* (intrastadial)	Horses	[13,14,15]
*A. cajennense*	Brazilian spotted fever (BSF)	*Rickettsia rickettsii*	Humans, capybaras	[16]
*A. sculptum*	Brazilian spotted fever (BSF)	*Rickettsia rickettsii*	Humans, capybaras	[17]
*A. variegatum*	Heartwater	*Ehrlichia ruminantium*	Domestic and wild ruminants	[18]
*Argas persicus*	Avian spirochetosis	*Borrelia anserine*	Turkeys, chickens, birds	[13]
*Dermacentor andersoni*	Tick paralysis	Tick proteins	Sheep, cattle, goats, other mammals, chickens	[13]
	Rocky mountain spotted fever	*Rickettsia rickettsii*	Wild-rodents, rabbits, humans	[19,20]
	Bovine anaplasmosis	*A. marginale*	Cattle, buffalo, large ruminants	[21]
*D. variabilis*	Bovine anaplasmosis	*A. marginale*	Cattle, buffalo, large ruminants	[21]
	Rocky mountain spotted fever	*Rickettsia rickettsii*	Wild-rodents, Opossums, humans	[19,20]
*Haemaphysalis longicornis*	Theileriosis	*Theileria orientalis*	Cattle, buffalo	[13]
*Ha. leachi*	Canine babesiosis	*Babesia canis*	Dogs	[22]
*Ha. spinigera*	Tropical theileriosis	*Theirleria annulata*	Cattle, buffalo	[13]
*Hyalomma anatolicum*	Ovine babesiosis	*Babesia* spp.	Sheep	[22,23]
	Equine babesiosis	*Babesia equi*	Horses	[24]
	Tropical theileriosis	*Theirleria annulata*	Cattle, buffalo	[25]
	Crimean-Congo hemorrhagic hever	Crimean-Congo Hemorrhagic Fever virus	Humans, goat, sheep, cattle	[26]
*Hy. marginatum*	Sweating sickness	Tick proteins	Cattle, sheep, other ruminants, dogs	[13]
	Crimean-Congo hemorrhagic fever	Crimean-Congo Hemorrhagic Fever virus	Humans, goat, sheep, cattle	[27]
*Hy. rufipes*	Crimean-Congo hemorrhagic fever	Crimean-Congo Hemorrhagic Fever virus	Humans, goat, sheep, cattle	[28]
*I. ricinus*	Tick-borne encephalitis	Flavivirus	Humans, rodents, insectivores	[13]
	Babesiosis (human babesiosis and redwater fever)	*Babesia microti*, *Babesia divergens*	Humans, cattle	[13]
*I. scapulariss*	Lyme borreliosis	*Borrelia burgdorferi* sensu stricto, *B. mayonii*	Humans, dogs, cats, rodents	[13]
	Anaplasmosis	*Anaplasma phagocytophilum*	Humans, rodents, cervids	[13]
	Babesiosis	*Babesia microti*, *B. odocoilei*	Humans, rodents, cervids	[13,29]
	Powassan virus lineage II (deer tick virus)	Flavivirus	Humans, rodents, insectivores	[13]
*Ornithodoros coriaceus*	African swine fever	Iridovirus	Domestic and wild pigs, warthogs	[30]
*O. lahorensis*	Tick toxicosis	Tick proteins	Cattle, sheep, birds	[13]
	African swine fever	Iridovirus	Domestic and wild pigs, warthogs	[13]
*O. porcinus*	African swine fever	Iridovirus	Domestic and wild pigs, warthogs	[13]
*Rhipicephalus appendiculatus*	East coast fever	Theileria parva	Cattle, buffalo	[13]
*R. (Boophilus) microplus*	Bovine babesiosis	*Babesia bovis, B. bigemina*	Cattle, water buffalo	[13]
	Heartwater	*E. ruminantium*	Domestic and wild ruminants	[31]
	Bovine anaplasmosis	*A. marginale*	Cattle, buffalo, large ruminants	[32]
*R. sanguineus*	Boutonneuse fever/Mediterranean spotted fever	*Rickettsia conorii*	Small mammals, hedgehogs, dogs, humans	[13]
	Rocky mountain spotted fever	*Rickettsia rickettsii*	Dogs, humans	[19,20]

* Previously identified as *Amblyomma cajennense*.

**Table 2 insects-11-00490-t002:** Tick resistance to synthetic acaricides.

Tick Species	Acaricide Class	Country Reported	Reference
*Amblyomma cajennense*	Pyrethroids	Brazil	[72]
*Amblyomma mixtum* *	Pyrethroids	Mexico	[73]
	Organophosphate	Mexico	[73,74]
*Hyalomma anatolicum*	Pyrethroids	India	[75]
	Organophosphate	India	[75]
*Rhipicephalus annulatus*	Macrocyclic lactones	Egypt	[76]
	Pyrethroids	Iran	[77]
*Rhipicephalus appendiculatus*	Pyrethroids	Uganda	[78]
	Organophosphate	Uganda	[78]
*Rhipicephalus bursa*	Organophosphate	Iran	[79]
*Rhipicephalus decoloratus*	Pyrethroids	Uganda	[78]
	Organophosphate	Uganda	[78]
*Rhipicephalus microplus*	Benzoylphenyl ureas	Brazil	[80]
	Formamidines	Australia, Zimbabwe	[81,82]
	Macrocyclic lactones	Mexico, Brazil, Colombia, Egypt	[76,82,83,84,85,86]
	Pyrethroids	Mexico, Brazil, Colombia, Argentina, US, Australia, India, New Caledonia (France territory), South Africa	[76,83,84,85,86,87,88,89,90,91,92,93,94,95]
	Organophosphate	Mexico, Argentina, Brazil, USA, Sri Lanka, India, Australia	[85,87,91,96,97,98,99]
	Organochlorine	Brazil	[86,100]
	Phenylpyrazole	Mexico, Brazil	[101]
*Rhipicephalus sanguineus*	Macrocyclic lactones	Mexico	[102]
	Pyrethroids	Brazil, Mexico, Panama, USA	[102,103,104]
	Organophosphate	Panama	[105]
	Phenylpyrazole	Brazil	[103,105]

* Previously identified as *Amblyomma cajennense*.

**Table 3 insects-11-00490-t003:** Plant-derived compounds with potential for tick control and prevention.

Class of Compound	Compound	Formula	Source	Effect	Species of Tick
Monoterpene	α-pinene	C_10_H_16_	*Plectranthus barbatus* *Rosmarinus officinalis* *Satureja myrtifolia*	acaricide	*R. microplus*
	β-pinene	C_10_H_16_	*Lindera melissifolia* *Stylosanthes humilis* *Cleome monophylla* *Clausena anisata* *Cannabis sativa*	repellent	*A. americanum* *R. appendiculatus*
	β-citronellol	C_10_H_20_O	*Pelargonium graveolens* *Dianthus caryophyllus*	acaricide, repellent	*A. americanum* *I. ricinus*
	Borneol	C_10_H_18_O	*Lavandula angustifolia* *Artemisia abrotanum* *Cunila spinate* *Origanum minutiflorum*	repellent	*H. marginatum* *I. ricinus* *R. turanicus*
	Carvacrol	C_10_H_14_O	*Chamaecyparis nootkatensis* *Gynandropsis gynandra* *Origanum minutiflorum* *Satureja thymbra* *Lippia gracilis*	acaricide	*H. marginatum* *I. Scapularis* *R. appendiculatus*
	Citronellal	C_10_H_18_O	*Cymbopogon nardus* *Corymbia citriodora* *Citrus hystrix*	acaricide	*A. cajennens* *D. nitens* *I. ricinus* *R. microplus*
	Elemol	C_15_H_26_O	*Maclura pomifera*	repellent	*A. americanum* *I. scapularis*
	Eucalyptol (1,8-cineole)	C_10_H_18_O	*Eupatorium adenophorum* *Lippia javanica* *Ocimum species*	acaricide	*H. longicornis* *H. marginatum* *R. microplus*
	Geraniol	C_10_H_18_O	*Pelargonium species* *Cymbopogon species* *Dianthus caryophyllus*	acaricide, repellent	*A. americanum* *A. cajennense* *I. ricinus* *R. microplus*
	Limonene	C_10_H_16_	*Citrus species* *Copaifera reticulata* *Hesperozygis ringens* *Tetradenia riparia*	acaricide	*R. microplus*
	Linalool	C_10_H_18_O	*Tagetes erecta* *Hesperozygis ringens* *Ocimum basilicum* *Origanum onites* *Cymbopogon martini*	acaricide	*H. bispinosa* *R. microplus* *R.turanicus*
	Myrcene	C_10_H_16_	*Origanum minutiflorum* *Lippia javanica* *Salvia nilotica*	acaricide	*H. marginatum* *R.turanicus*
	Pulegone	C_10_H_16_O	*Mentha suaveolens*	acaricide	*H. aegyptium*
	Tagetone	C_10_H_16_O	*Tagetes species*	acaricide	*H. bispinosa* *H. marginatum* *R. sanguineus*
	Thymol	C_10_H_14_O	*Thymus vulgaris* *Lippia sidoides* *Lippia gracilis* *Origanum minutiflorum*	acaricide	*A. cajennense* *R. sanguineus* *R. turanicus*
Diterpene	Callicarpenal	C_16_H_26_O	*Callicarpa americana*	acaricide, repellent	*A. cajennense*
Fatty acid amide	Spilanthol	C_14_H_23_NO	*Acmella Oleracea*	acaricide	*R. microplus,* *D. nitens*
Sesquiterpene	α-humulene	C_15_H_24_	*Lindera melissifolia* *Stylosanthes humilis* *Cleome monophyla*	repellent	*R. appendiculatus*
	β-caryophyllene	C_15_H_24_	*Syzygium aromaticum* *Cannabis sativa*		*I. ricinus* *R. microplus*
	Nootkatone	C_15_H_22_O	*Chamaecyparis nootkatensis* *Chrysopogon zizanioides* *Citrus grandis*	acaricide	*I. scapularis*
Tetranotriterpenoid	Azadirachtin	C_35_H_14_0_16_	*Azadirachta indica* *Melia azedarach*	acaricide	*A. cajennense* *R. microplus*
Naphthoquinone	Plumbagin	C_11_H_8_O_3_	*Plumbago zeylanica*	acaricide	*A. variegatum*
Organosulfur	Allicin	C_6_H_10_OS_2_	*Allium sativum*	acaricide, repellent	*H. marginatum* *R. microplus*
Phenylpropanoid	Eugenol	C_10_H_12_O_2_	*Ocimum species* *Artemisia species* *Plectranthus barbatus*	acaricide	*H. anatolicum* *I. ricinus* *R. appendiculatus* *R. microplus* *R. sanguineus*
Pyrethrin	Pyrethrin I	C_21_H_28_O_3_	*Chrysanthemum species*	acaricide	*D. reticulatus* *D. variabilis* *I. scapularis* *R. sanguineus*
Resin	Oleoresin	C_18_H_27_NO_3_	*Copaifera reticulata*	acaricide	*R. microplus*
Steroidal glycoside	Digitoxin	C_41_H_64_O_13_	*Calotropis procera* *Digitalis purpurea*	acaricide	*H. dromedarii* *R. microplus*

Source: Adapted from [114], complemented by [120,125,126,127,128].

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
