# Peer review of "Plant-Derived Natural Compounds for Tick Pest Control in Livestock and Wildlife: Pragmatism or Utopia?"

_insects, 2020, doi:10.3390/insects11080490_

Round 1

Reviewer 1 Report

Attached is my review of the manuscript.
The manuscript is acceptable with minor changes.

Author Response

Reviewer 1

  1. Section 2 contains a statement that 80% of tick borne pathogens have a wildlife connection. It is probably closer to 100%. Off-hand I can’t think of any that do not have either a reservoir in nature or that are not susceptible because they are closely related to pets or livestock.

We agree with reviewers 1 and now the statement reads “As nearly 100% of the tick-borne pathogens present in the United States have a potential wildlife host component, a need remains to provide new economically feasible tools to reduce contact at the livestock–wildlife interface [33].”

  1. Section 3, statement: “strategic tick control based on tick-population dynamics” requires clarification. Do you mean timing of applications or targeting of applications?

We have changed that statement and now they read “IPM based on tick abundance” to clarify that the management strategy is based on tick abundance.

  1. End of section 3, suggest change of wording from “exploitations” to “production”

We concur with the reviewer and have changed the wording. The sentences now read “Organic livestock systems represent only 0.7 and 3.3% of pigs and poultry production, respectively. Whereas sheep and bovines represent just 5.0 and 5.2% of the organic livestock production, respectively [110].”

  1. Section 4, paragraphs on neem. The scientific name Azadirachta should be moved up to the first paragraph where neem is first mentioned. Also the second paragraph has only one sentence so it should be combined with the first.

  1. The paragraphs on neem are poorly written and need a rewrite.

We have moved the scientific name and have re-written the paragraph as follow: “Neem (Azadirachta indica) has also been tested as acaricide. Aqueous extracts and oils have been tested due to their potential use as insecticides [143] and acaricides [144-146].  Azadirachtin is the most widely studied component purified from neem oils. A study comparing the effects of different concetrations of Azadirachtin and neem leaf extracts on R. sanguineus larvae showed 80% and 95% mortalities after the Larval Packet Test (LPT), respectively, in the lower concentrations (0.5% and 10%, respectively) [146]. Nevertheless, this acaricidal effect was diminished in experiments with higher concentrations [146]. In adult females, treatment with the aqueous extracts from neem leaf at 10% and 20% results in morphological changes during oocyte development when compared to control samples [145]. Similar outcomes were reported after treatment with neem oils containing Azadirachtin [147], indicating that that the changes in morphology are possibly connected with this compound. Furthermore, neem oil also reduces cuticle thickness and distorts epithelial cell morphology of semi-engorged females [144]. However, neem leaf extracts can negatively affect oocytes and ovaries in mammals. Studies with rats indicate that neem leaf extracts increase oxidation in the oocytes and lead to apoptosis [148,149]. Although neem leaf material can produce anemia, reduced fertility, and cause abortions, aqueous extracts and purified components appear to be less toxic and require high concentrations to have  negative effects on mammals [143]; supporting the potential of this plant as source for acaricidal compounds.”

  1. End of section 4.1, italicize, plant names, tick names and “in vitro”.

We have italicized the scientific names and “in vitro” as requested by the reviewer. The text now reads: “More recently, six medicinal plants (Vernonia amygdalina, Calpurnia aurea, Schinus molle, Ricinus communis, Croton macrostachyus, and Nicotiana tabacum) were used against R. decoloratus and R. pulchellus in an in vitro adult immersion test with five crude extract concentrations (6.25, 12.5, 25, 50, or 100 mg/ml) with diazinon (0.1%) as a positive control. Two of the concentrations (50, and 100 mg/ml) showed a comparable and strong acaricidal effect on both tick species compared with the positive control after a 24 hr period [140].”

  1. Section 4.1. Many of the plants under discussion are poisonous to livestock having compounds that are not tannins. The authors need to explain why they think the tannins and not these poisons are active against ticks.

We concur with the reviewer. We did not mention the negative effects of some of the plants. We have added several clarifications throughout the manuscript, such as “Although have been used dating back to ancient times, some medicinal plants identified in ethnoveterinary and even in human medicine can have toxic properties depending on their origin and nature [125,126]. However, toxic plants may contain active compounds with useful biological activities for biomedical applications [127,128]. For instance, glycosides, alkaloids, saponins, tannins, volatile oils, flavonoids, and diterpenoids are examples of active components that can be potentially toxic but are used in ethnoveterinary [129,130]. Obviously, the intercalating concern between pharmacology and toxicology is dose-dependent [127]. It is essential to be aware of the toxicity that plants of veterinary significance can have to avoid disease or mortality in livestock [131,132]. Studies defining the concentration and dosage of a component extract that leads to detrimental effects in animals and humans are needed to define working dosages.”, “However, neem leaf extracts can negatively affect oocytes and ovaries in mammals. Studies with rats indicate that neem leaf extracts increase oxidation in the oocytes and lead to apoptosis [148,149]. Although neem leaf material can produce anemia, reduced fertility, and cause abortions, aqueous extracts and purified components appear to be less toxic and require high concentrations to have  negative effects on mammals [143]; supporting the potential of this plant as source for acaricidal compounds.” And “Generally considered safe for mammals, some plant-derived products have been shown to exert negative health and welfare effects in humans and other animals [214]. Therefore, regulations requiring ecotoxicity studies, possible collateral effects, and education about how to use these compounds and the risks that they impose may minimize the possible negative impacts.”

  1. Section 4.3. What is meant by “virulence of ticks.”?

  1. Section 4.3 has a sentence that unspecified “others” react with heme and alter cell structure. This concept needs to be fleshed out. What are these “others” and what cells are they altering?

After revising that section, we also find that it is confusing. We have therefore deleted those sentences and the section now reads “The mode of action of many plant-derived compounds used for tick control is not completely understood. Some essential oils cause neurotoxic effects, such as inhibition of acetylcholinesterase (AChE), antagonism with the receptors of the octopamine neurotransmitter, or closure of the chloride channels by gamma-aminobutyric acid (GABA) [169]. Likewise, the exact mode of action of many plant essential oils has not been elucidated, a few studies have been conducted to understand how these naturally-occurring compounds act on ticks. Gross et al. [170] developed an in vitro assay, using hamster cell lines (CHO cells) expressing the R. microplus Tyramine receptor RmTAR1e, to screen components from essential oils that interact with this receptor. Using this assay, they identified a Tyramine receptor as the potential target for pulegone, carvacrol, isoeugenol, 1,4-cineole, and piperonyl alcohol. However, how other components and essential oils affect tick biology remains unexplored and warrants further investigation.”

Reviewer 2 Report

Dear Authors,  This is very fine review and seems very polished.  I don't have any edits to the text.

In references, you are missing the citation for Goolsby et al. 2018.

One substantive comment I think needs to be discussed about botanical acaricides.  You may be able to use them on livestock that are held in a chute or other device that restricts movement, but many botanicals elicit a strong negative response from the animal.  The botanicals may smell nice to us, but seem to be repellant and offensive to livestock.  I think the same will be true with wildlife.  Wildlife are very sensitive to odors and plant derived compounds can be very strong smelling.  

Author Response

Reviewer 2

In references, you are missing the citation for Goolsby et al. 2018.

We have added the references to Dr. Goolsby work on nematodes as follow: “6.    Singh, N.; Goolsby, J.; Shapiro Ilan, D.I.; Miller, R.; Thomas, D.B.; Klafke, G.; Tidwell, J.P.; Racelis, A.; Grewal, P.; Perez De Leon, A.A. Efficacy of entomopathogenic nematodes (Rhabditida: Heterorhabditidae and Steinernematidae) against engorged females of the cattle fever tick Rhipicephalus (Boophilus) microplus (Acari: Ixodidae). Southwestern Entomologist 2018, 43, 1-17, doi:https://doi.org/10.3958/059.043.0119.

  1. Goolsby, J.; Singh, N.K.; Shapiro, D.I.; Miller, R.; A.A., P.D.L. Comparative efficacy of entomopathogenic nematodes against multi-acaricide resistant strain of cattle fever tick, Rhipicephalus microplus (Acari: Ixodidae). Southwestern Entomologist. 2019, 44, 143-153, doi:https://doi.org/10.3958/059.044.0116.”

One substantive comment I think needs to be discussed about botanical acaricides.  You may be able to use them on livestock that are held in a chute or other device that restricts movement, but many botanicals elicit a strong negative response from the animal.  The botanicals may smell nice to us, but seem to be repellant and offensive to livestock.  I think the same will be true with wildlife.  Wildlife are very sensitive to odors and plant derived compounds can be very strong smelling.  

We agree with Reviewer 2 and Reviewer 1 in these points.

We have added several clarifications throughout the manuscript, such as “Although have been used dating back to ancient times, some medicinal plants identified in ethnoveterinary and even in human medicine can have toxic properties depending on their origin and nature [125,126]. However, toxic plants may contain active compounds with useful biological activities for biomedical applications [127,128]. For instance, glycosides, alkaloids, saponins, tannins, volatile oils, flavonoids, and diterpenoids are examples of active components that can be potentially toxic but are used in ethnoveterinary [129,130]. Obviously, the intercalating concern between pharmacology and toxicology is dose-dependent [127]. It is essential to be aware of the toxicity that plants of veterinary significance can have to avoid disease or mortality in livestock [131,132]. Studies defining the concentration and dosage of a component extract that leads to detrimental effects in animals and humans are needed to define working dosages.”, “However, neem leaf extracts can negatively affect oocytes and ovaries in mammals. Studies with rats indicate that neem leaf extracts increase oxidation in the oocytes and lead to apoptosis [148,149]. Although neem leaf material can produce anemia, reduced fertility, and cause abortions, aqueous extracts and purified components appear to be less toxic and require high concentrations to have  negative effects on mammals [143]; supporting the potential of this plant as source for acaricidal compounds.” And “Generally considered safe for mammals, some plant-derived products have been shown to exert negative health and welfare effects in humans and other animals [214]. Therefore, regulations requiring ecotoxicity studies, possible collateral effects, and education about how to use these compounds and the risks that they impose may minimize the possible negative impacts.”